Genome-wide investigation of the AP2/ERF superfamily and their expression under salt stress in Chinese willow (Salix matsudana)

Zhang Jian 1
Shi Shi zheng 2
Jiang Yuna 1
Zhong Fei 1
Liu Guoyuan 1
Yu Chunmei 1
Lian Bolin 1
Chen Yanhong chenyh@ntu.edu.cn 1
1 Lab of Landscape Plant Genetics and Breeding, School of Life Science, Nantong University , Nantong , Jiangsu , China
2 Jiangsu Academy of Forestry , Nanjing , Jiangsu , China
Sun Genlou
Electronic publication date: 2021 Apr 13
Publication date: 2021
Volume: 9
Electronic Location ID: e11076
Received 2020 Oct 14; Accepted 2021 Feb 17
Copyright: ©2021 Zhang et al.
Copyright year: 2021
Copyright holder: Zhang et al.
License: This is an open access article distributed under the terms of the Creative Commons Attribution License, which permits unrestricted use, distribution, reproduction and adaptation in any medium and for any purpose provided that it is properly attributed. For attribution, the original author(s), title, publication source (PeerJ) and either DOI or URL of the article must be cited.
License URL: https://creativecommons.org/licenses/by/4.0/

Keywords: Salix matsudana, Salt tolerance, Expression patterns, Syntenic relationship, Evolution, Phylogenetic analysis, Duplication, SmAP2/ERF family protein

Funding: The National Natural Science Foundation of China 31971681 Jiangsu Province Forestry Science and Technology Innovation and Promotion Project LYKJ [2018]36 Nantong University Scientific Research Start-up project for Introducing Talents 18R08 This work was supported by the National Natural Science Foundation of China (31971681), Jiangsu Province Forestry Science and Technology Innovation and Promotion Project (LYKJ [2018]36) and the Nantong University Scientific Research Start-up project for Introducing Talents (18R08). The funders had no role in study design, data collection and analysis, decision to publish, or preparation of the manuscript.

==============================
AP2/ERF transcription factors (TFs) play indispensable roles in plant growth, development, and especially in various abiotic stresses responses. The AP2/ERF TF family has been discovered and classified in more than 50 species. However, little is known about the AP2/ERF gene family of Chinese willow (Salix matsudana), which is a tetraploid ornamental tree species that is widely planted and is also considered as a species that can improve the soil salinity of coastal beaches. In this study, 364 AP2/ERF genes of Salix matsudana (SmAP2/ERF) were identified depending on the recently produced whole genome sequencing data of Salix matsudana. These genes were renamed according to the chromosomal location of the SmAP2/ERF genes. The SmAP2/ERF genes included three major subfamilies: AP2 (55 members), ERF (301 members), and RAV (six members) and two Soloist genes. Genes’ structure and conserved motifs were analyzed in SmAP2/ERF family members, and introns were not found in most genes of the ERF subfamily, some unique motifs were found to be important for the function of SmAP2/ERF genes. Syntenic relationships between the SmAP2/ERF genes and AP2/ERF genes from Populus trichocarpa and Salix purpurea showed that Salix matsudana is genetically more closely related to Populus trichocarpa than to Salix purpurea. Evolution analysis on paralog gene pairs suggested that progenitor of S. matsudana originated from hybridization between two different diploid salix germplasms and underwent genome duplication not more than 10 Mya. RNA sequencing results demonstrated the differential expression patterns of some SmAP2/ERF genes under salt stress and this information can help reveal the mechanism of salt tolerance regulation in Salix matsudana.

Introduction

APETALA 2/ethylene-responsive element binding factors (AP2/ERF) are important transcription factors (TFs) coded by genes from the AP2/ERF superfamily. All of the members of this superfamily possess AP2 domains and, according to the number and structure of AP2 domains, the superfamily is divided into several categories, including AP2, ERF, RAV, and Soloist (Nakano et al., 2006). Most of the AP2 gene family members have two AP2 domains and can be further divided into AP2 and ANT groups; ERF family members have only one AP2 domain and can also be subdivided into ERF and DREB subfamilies based on binding motifs in the promoter of downstream genes. Members of the ERF and DREB subfamilies are classified into 12 groups (groups A1–B6). DREB includes groups A1–A6, whereas ERF includes groups B1–B6 (Nakano et al., 2006). In addition to one AP2 domain, RAVs also have one B3 domain. The Soloist group contains a single AP2 domain with sequence divergence from the AP2 and ERF families and has less than three members in most species (Song et al., 2016).

The AP2/ERF superfamily is plant-specific and has more than 100 members in many plant species; for example, there are 147 members in Arabidopsis, 200 members in Populus trichocarpa, and more than 500 members in the tetraploid crop Brassica napus (Nakano et al., 2006; Song et al., 2016; Zhuang et al., 2008). Different members play various regulatory roles in plant growth and development, defense response, fruit ripening, and metabolism (Gu et al., 2017). Several recent reports demonstrated functions of AP2/ERF2 TFs in plant development. For example, loss of DRNL function affects gynoecium development (Duran-Medina et al., 2017); the function of Populus ERF139 (Potri.013G101100) in xylem cell expansion was characterized by transgenic overexpression and dominant repressor lines of ERF139 (Wessels et al., 2019); RhERF1 and RhERF4 play roles in petal abscission in rose (Gao et al., 2019); and a maize AP2/ERF TF, ZmRAP2.7, is involved in brace root development. AP2/ERF TFs such as ZmEREB94 and CitAP2.10 also play important roles in plant metabolism; ZmEREB94 acts as a key regulator of starch synthesis in maize (Li et al., 2017b), and CitAP2.10 was characterized as a regulator of (+)-valencene synthesis in sweet orange fruit (Shen et al., 2016).

The AP2/ERF superfamily plays major and crucial roles in abiotic stress tolerance, which is why this superfamily has received special attention by plant scientists (Gu et al., 2017; Srivastava & Kumar, 2018). Through extensive investigation on their regulatory mechanism, people want to elucidate their potential applications in crop improvement (Srivastava & Kumar, 2018). Members of this superfamily (primarily ERFs and DREBs) have been prominently used to improve stress tolerance in plants. To improve salinity stress tolerance, many genes from different species were identified. IbRAP2-12, an AP2/ERF gene cloned from the salt-tolerant sweet potato, and LkERF-B2 from Larix kaempferi promotes tolerance to salt and drought stresses in overexpressing Arabidopsis lines (Cao, Shu & Li, 2019; Li et al., 2019b). Overexpression of HARDY, an AP2/ERF gene from Arabidopsis, improves drought and salt tolerance by reducing transpiration and sodium uptake in transgenic Trifolium al.exandrinum L (Abogadallah et al., 2011). A soybean DREB ortholog, GmDREB1, enhances the salt tolerance in transgenic alfalfa (Jin et al., 2010).

Comparative genomic analysis of model plants such as Arabidopsis have provided unprecedented advantages for gene discovery and functional annotation of newly sequenced plant genomes (Brendel, Kurtz & Walbot, 2002; Hall, Fiebig & Preuss, 2002; Schranz et al., 2007). By exploring the available genomic data, AP2/ERF gene families from 50 species were discovered and classified, and provide critical guidance for functional analysis (Srivastava & Kumar, 2018). For example, in radish, cauliflower, and celery, whole genome identification and classification of AP2/ERF gene family members were carried out; additionally, expression patterns of different members under different stresses were revealed, and the function of candidate genes was verified (Karanja et al., 2019; Li et al., 2017a; Li et al., 2019a).

Salix matsudana Koidz., an allotetraploid member of Salicaceae, is an important ornamental tree species native to northeastern China (Zhang et al., 2016; Zhang et al., 2017; Zhang et al., 2020); it is widely cultivated and considered an important economic plant because of its easy vegetative propagation, rapid growth, and substantial biomass yields. Salix matsudana also plays an important ecological role when grown along Chinese coastal beaches, where the salinity content is high (Zhang et al., 2016). This species can improve the beach soil and alleviate salinization. Newly reclaimed beach soil has higher salinity and requires new germplasm with higher salinity tolerance (Zhang et al., 2017). Because the AP2/ERF gene family members have regulatory roles in salinity tolerance, whole genome characterization of the AP2/ERF gene family in Salix matsudana will reveal mechanisms underlying stress signal transmission and provide guidance for selection or creation of new germplasm with higher salinity tolerance. In total, 200 and 173 AP2/ERF superfamily genes were identified from two species, diploid salix germplasm Salix arbutifolia and Populus trichocarpa, respectively (Rao et al., 2015; Zhuang et al., 2008). The Salix matsudana genome was recently sequenced and assembled (Zhang et al., 2020); as an allotetraploid, identification of the AP2/ERF gene family will reveal the evolutionary relationship with poplar and other members of Salix, and the molecular mechanisms responsible for salinity stress responses.

Material and Methods

Plant material and salt stress treatment

The salinity stress treatments were carried out on Salix matsudana ‘yanjiang’ and Salix matsudana ‘9901’. Our previous research results showed that ‘yanjiang’ was salt-sensitive variety, while ‘9901’ was salt-tolerant variety (Zhang et al., 2016). The two salix cultivars used in this study were collected from the botany garden of Nantong University (Nantong, China). The two salix cultivars were authorized for only scientific research purpose, and were deposited in school of life science in Nantong University. The stem cuttings (length, 8–10 cm; coarse, 2–3 mm) of two materials were selected for hydroponic rooting for 20 days. The stems with new generated roots were dipped into NaCl solution (150 mM) (only root and part of shoots were immersed in the solution) for 4 h. All root samples were divided into four categories with three biological replicate to do RNA sequencing: Sample1-1/Sample1-2/Sample1-3 (‘yanjiang’ without salt stress treatment, YJ CK), Sample1N-1/Sample1N-2/Sample1N-3 (‘yanjiang’ with salt stress treatment, YJ NT), Sample2-1/Sample2-2/Sample2-3 (‘9901’ without salt stress treatment, 9901 CK), and Sample2N-1/Sample2N-2/Sample2N-3 (‘9901’ with salt stress treatment, 9901 NT).

RNA isolation and Real-time Quantitative PCR (qRT-PCR) analysis

Total RNA was extracted using TaKaRa MiniBEST Plant RNA Extraction Kit (Takara, Dalian, China) from roots according to the manufacturer’s instruction. Four samples (Yanjiang, Yanjiang NT, 9901, 9901 NT) were collected following the same samples treatments procedure as that in RNA sequencing. For each sample, 3 µg of total RNA was used to synthesize first-strand cDNA with SuperScriptII reverse transcriptase (Takara, Dalian, China). For qRT-PCR, the reaction preparation, application parameter settings and quantitative analysis were performed as previously described (Chen et al., 2018). The reactions were performed using the ABI Prism 7000 Real-time PCR system (Applied Biosystems, USA). The Salix purpurea Actin1 gene (SapurV1A.0655s0050.1) were used as reference genes. The gene-specific primers for the 15 selected genes are listed in Table S1.

Genome sequence retrieval

The Populus trichocarpa and Salix purpurea sequences were downloaded from JGI (http://www.phytozome.net/). The Salix matsudana sequences were obtained from our sequencing, and assembly results were obtained by Roche/454 and Illumina/HiSeq-2000 sequencing technologies (Zhang et al., 2020).

Identification of AP2/ERF genes in Salix matsudana and Salix purpurea

The Pfam accession number of AP2 domain is PF00847 (Gathering cut-off value, 20.6). We downloaded the Hidden Markov Model (HMM) profile for the AP2/ERF TFs from the Pfam database (http://pfam.xfam.org/) with Pfam accession number PF00847 as the search keyword. An alternative HMM profile was built by sequence alignment using ClustalW (Larkin et al., 2007). Two HMM profile files were provided as File S1 and File S2. Using an in-house Perl script with two HMM profiles as queries, hmmsearch was carried out by searching the Salix matsudana and Salix purpurea protein databases with default parameters. To validate the putative accuracy of two HMM search results, the candidate protein sequences were checked in three websites: SMART (http://smart.embl.de/#), CDD (https://www.ncbi.nlm.nih.gov/cdd/), and Pfam (http://pfam.xfam.org/). Candidate proteins with positive results from all three websites were selected as AP2/ERF family members of Salix matsudana and Salix purpurea. Additionally, putative AP2/ERF protein characteristics, including length, molecular weight, and isoelectric point, were calculated by the ExPasy site (http://au.expasy.org/tools/pi_tool.html). The genes CDS sequences were listed in File S3.

Phylogenetic analysis and classification of SmAP2/ERF genes

Using an in-house Perl script (domain_xulie.pl), the conserved AP2 core domains of putative SmAP2/ERF proteins were obtained and subjected to multiple sequence alignment using ClustalW (Larkin et al., 2007). To better classify these SmAP2/ERF proteins, 48 AP2 domains from known categories of Arabidopsis and Populus trichocarpa AP2 genes were selected to carry out multiple sequence alignment with SmAP2/ERF proteins, and a phylogenetic tree based on this alignment was built by MEGA 7.0 with the neighbor-joining method with default parameters (Kumar, Stecher & Tamura, 2016). Bootstrap value was set to 1,000. Depending on the phylogenetic tree constructed by SmAP2, PtAP2, and AtAP2 domains, these SmAP2/ERF proteins were classified into different subfamilies and subgroups.

Gene structure and conserved motif structure analysis

The UTR–exon–intron structures of the SmAP2/ERF genes were obtained based on the gene annotation gff3 files we assembled. Using the online website tool Gene Structure Display Server (http://gsds.cbi.pku.edu.cn/), we obtained the gene structure diagrams (Hu et al., 2015).

To characterize the structures of SmAP2/ERF proteins, the online tool MEME (http://meme-suite.org/tools/meme) was used to search for conserved motifs (Bailey et al., 2009). The optimized parameters were employed as follows: any number of repetitions, maximum number of motifs = 10, and the optimum width of each motif was 6–50 residues. The search result file meme.xml was downloaded from the website and opened by TBtools v0.66831 to obtain the gene structure diagram (Chen et al., 2020a; Chen et al., 2020b).

Gene position on chromosomes, and gene tandem and segmental duplication analysis

Using the “Amazing Gene Location from GFF3/GTF File” tool of TBtools, the SmAP2/ERF genes were mapped on 38 chromosomes of Salix matsudana. Because some scaffolds were not assembled onto the chromosomes, not all SmAP2/ERF genes mapped onto the chromosomes (Chen et al., 2020a; Chen et al., 2020b).

Salix matsudana is a tetraploid willow. Gene duplication events are a common phenomenon in the genome. There are two kinds of gene duplications in the genome: tandem duplication events (TDs) and segmental duplication events (SDs). TDs refer to two or more adjacent homologous genes located within 200 Kb on a single chromosome; SDs refer to homologous gene pairs between different chromosomes (Cannon et al., 2004). The gene duplication pairs were identified in TBtools by the “Blast compare 2 Seq [sets] <Big File>” and “Quick McscanX Wrapper” tools. The candidate duplicated genes should have ≥80% coverage and ≥65% similarity. The TDs of SmAP2 genes were revealed on a chromosome using the “Amazing Gene Location from GFF3/GTF File” tool of TBtools. The SDs of SmAP2 genes were visualized by the “Amazing Super Circos” tool of TBtools (Chen et al., 2020a; Chen et al., 2020b).

Divergence time calculation of duplicated genes

After BLASTn analysis of CDS sequences and obtaining duplicated gene pairs, the nonsynonymous substitution rate (Ka) and Synonymous substitution rate (Ks) of gene pairs were calculated by the “Simple Ka/Ks calculator (NG)” tool of TBtools. The divergence time was estimated with the formula: T = Ks/2 λ (Song et al., 2016). The clock-like rate λ value (9. 1 ×10−9) from Populus was used in the calculation (Lei et al., 2012; Lynch & Conery, 2000).

Collinearity analysis between Salix matsudana and the representative species

To demonstrate the syntenic relationships of the orthologous SmAP2/ERF genes obtained from Salix matsudana and other two selected plants (Populus trichocarpa, and Salix purpurea), the syntenic analysis maps were constructed using the “Amazing Super Circos” tool of TBtools (Chen et al., 2020a; Chen et al., 2020b).

RNA sequencing and a heat map generated by hierarchical clustering

Transcriptome sequencing data of 12 samples were obtained by Illumina HiSeq sequencing. Using TopHat2 software (Kim et al., 2013), the clean reads were mapped to the reference genome sequence of S. matsudana. Gene expression levels were estimated by fragments per kilobase of transcript per million fragments mapped (FPKM) (Jin, Wan & Liu, 2017). The FPKM values of all genes from RNA sequencing were available as File S4. Differential expression analysis of two conditions/groups was performed using the DESeq R package (1.10.1). To identify DEGs, fold change ≥ 2 and false discovery rate (FDR) < 0.01 were used as screening criteria. Using the “Amazing HeatMap” tool of TBtools, a graph of the expression level of SmAP2/ERF family genes with hierarchical clustering was generated (Chen et al., 2020a; Chen et al., 2020b).

Results

Identification, phylogenetic analysis, and classification of 364 AP2/ERF TF family members in Salix matsudana

By HMM profile search against the Salix matsudana protein database, a total of 364 full-length AP2/ERF family proteins containing at least one AP2/ERF domain were identified as AP2/ERF superfamily members of Salix matsudana (Fig. 1). The original hmmsearch –domtblout results are listed in Fil S5. The name, protein length, molecular weight, and isoelectric point of individual genes are listed in Table S2.

Figure 1 Unrooted phylogenetic tree and classification of 364 SmAP2/ERF genes and their representative orthologs from Arabidopsis and Populus.

The amino acid sequences of AP2 domains from 364 SmAP2/ERF proteins and 48 orthologs from Arabidopsis and Populus were aligned by ClustalW, and the neighbor-joining tree was constructed using MEGA 7.0 with 1,000 bootstrap replicates. The evolutionary distances were computed using the p-distance method. In total, 364 SmAP2/ERF members were classified into 15 smaller subgroups, and their names are labeled beside the tree.

The phylogenetic relationships of SmAP2/ERF proteins were inferred by multiple sequence alignment of the AP2 domain, which included approximately 50–60 amino acids. The sequence alignment of all AP2/ERF genes showed some conserved amino acids at specific positions, as previously reported (Nakano et al., 2006) (Fig. S1). For example, the WLG element (58th–60th amino acids; 58–60AA) was highly conserved in the ERF and RAV families; alternatively, the conserved sequences from 58–60AA were converted into YLG elements in the AP2 family and HLG element in two sololist members (Liu et al., 2019). In many species, these conserved amino acid profiles contribute to convincing classification of AP2/ERF genes. Based on multiple sequence alignments of 48 AP2/ERF proteins from Arabidopsis and Populus trichocarpa with known categories and 364 Salix matsudana AP2/ERF proteins, we constructed a phylogenetic tree using the neighbor-joining method to explore the phylogenetic relationships of Salix matsudana AP2/ERF proteins. The phylogenetic tree showed that there were 55 AP2/ERF genes that belong to the AP2 family, with 47 genes that encode proteins with two AP2 domains and eight genes (SmAP2-20, SmAP2-25, SmAP2-29, SmAP2-35, SmAP2-36, SmAP2-40, SmAP2-41 and SmAP2-55) that encode proteins with a single AP2 domain (Fig. 1). Additionally, 301 genes that were predicted to encode proteins with a single AP2 domain were members of the ERF family. The ERF family could be further classified into two subfamilies, ERF and DREB. Of the 301 members, 166 and 135 genes belonged to the ERF and DREB subfamilies, respectively. The ERF family genes from Salix matsudana were distributed in B1–B6 subgroups; the DREB family genes from Salix matsudana were classified into A1–A6 subgroups. The gene number and percentage of each subgroup are listed in Fig. 2 and Table S3. Six putative genes were classified as RAV subgroup genes that encode proteins containing one AP2/ERF domain and one B3 domain (Fig. 1). Two genes were designated as Soloist genes, whose AP2/ERF-like domain sequences had lower homology compared with other AP2/ERF genes (Fig. 1).

Figure 2 Classification and subgroup proportions of SmAP2/ERF family genes.

The size of each piece is proportional to the relative abundance to the SmAP2/ERF genes assigned to this group.

The AP2/ERF genes number, classification and percentage of different subgroups from five plant species, including the model plant Arabidopsis, Populus, and two Salix plants, are listed in Table S3. The gene name of AP2/ERF genes from Populus trichocarpa and Salix purpurea are listed in Table S4. As a tetraploid plant, the total number (364) of AP2/ERF genes was much larger in Salix matsudana than in the other four species. The number of AP2/ERF genes in Salix matsudana was 2.5-, 1.8-, 1.9-, and 2.1-fold higher than those in A. thaliana (Nakano et al., 2006), Populus trichocarpa (Zhuang et al., 2008), Salix purpurea, and Salix arbutifolia (Rao et al., 2015, respectively. For DREB and ERF subfamilies, the percentage of all AP2/ERF genes in Salix matsudana was similar to those of A. thaliana, Populus trichocarpa, and Salix purpurea, and the percentages of DREB and ERF subfamilies were 38% and 45%, respectively. In Salix arbutifolia, the percentage of DREB (33%) was lower than that of the other four species, whereas the percentage of ERF (50.8%) was higher. In Salix matsudana, the percentage of the AP2 subgroup was highest among all five species (15%) and the numbers of most of gene sub-classifications were doubled, including the Soloist gene; there were two Soloist genes in the Salix matsudana genome. However, no duplications were observed in the RAV subgroup, and only six RAV genes were found in the Salix matsudana genome.

Figure 3 Phylogenetic relationships, gene structure, and architecture of conserved protein motifs in SmAP2/ERF superfamily members.

(A) The phylogenetic tree was constructed based on the amino acid sequences of the AP2 domain from 364 SmAP2/ERF proteins using MEGA7.0. The subgroup members was labeled by different colour and abbreviation name of subgroup. B1-B6 represented six ERF subgroups; A1-A6 represented six DREB subgroups; RAV represented six RAV members; Sol represented two sololist members; AP2 represented AP2 subfamily. (B) Motif composition of SmAP2/ERF proteins. Motifs 1–10 are displayed in different colored boxes. The sequence information for each motif is provided in Fig. S2. (C) Exon/intron structure of SmAP2/ERF genes. Yellow boxes indicate untranslated 5′- and 3′-regions; green boxes indicate exons; black lines indicate introns. The protein and gene length can be estimated using the scale at the bottom of B and C, respectively.

Gene structure and conserved motif analysis

To understand the structural diversity of SmAP2/ERF genes in different clades, a different form of phylogenetic tree of SmAP2/ERF family was constructed and the different subgroups were labled (Fig. 3A). The intron and exon structures of SmAP2/ERF genes were revealed by inputting Gff3 files into TBtools (Fig. 3B). A total of 55 genes of the AP2 subfamily had more exons than ERF and other subfamilies. Apart from three exons in the SmAP2-29 and four exons in the SmAP2-20, other members of the AP2 subfamily contained more than seven exons. The intron number was less than three in many members of the ERF and RAV subfamilies. In total, 215 gene members did not have introns (Fig. 3B). The exon/intron structures of genes that were classified in the same clade were similar. Many gene pairs were found in the phylogenetic tree that potentially originated from allotetraploid evolution of Salix matsudana. Many gene pairs (approximately 70%) maintained the same or similar gene structure during Salix matsudana evolution, which indicated that the SmAP2/ERF genes were conserved at the DNA level after polyploidization.

TF proteins always contain many conserved motifs to activate gene expression. A total of 10 conserved motifs were detected in 364 SmAP2/ERF proteins using the online MEME software, and a block diagram was constructed to characterize SmAP2/ERF protein structure (Fig. 3C, Fig. S2). Motif-4, Motif-1, Motif-2, Motif-3, Motif-5, Motif-7, and Motif-9 were found in the AP2 domain regions. The Motif-5 region covered the region of Motif-4 and Motif-1, whereas Motif-7 included Motif2 and Motif3. Motif-9 is a specific motif that is only found in the second AP2 domain of the AP2 subgroup. Motif-1, Motif-2, Motif-3, and Motif-4 were detected in 90% percent of the ERF subfamily proteins. Thirty proteins of the ERF subfamily lacked one or two motifs of Motif-1–4. An extreme example is SmERF B2-13, which only had Motif-2. Motif-6, Motif-8, and Motif-10 are motifs located outside of the AP2 domain. Motif-6 was primarily found in the AP2 subfamily with only one exception, SmERF B4-4, which was in the ERF-B4 clade. In the AP2 subfamily, members with two AP2 domains had Motif-6 located between the two AP2 domains. Motif-8 was found in 69 proteins of the AP2/ERF family, and its location was adjacent to the carboxyl terminal of Motif-3. Many proteins from the DREB-A1, DREB-A4, DREB-A5 clades had Motif-8. Motif-10 was found in 62 proteins of the AP2/ERF family, with 61 proteins from the ERF subfamily and only one from the AP2 subfamily. Motif-10 was mostly distributed on the proteins from the ERF-B3, DREB-A2, and DREB-A4 clades. The functions of these three motifs need to be elucidated by further experimental analysis.

Besides protein SmAP2-20, the entire AP2 domain was distributed in the amino terminal or in the middle position of the proteins. In the two Soloist genes, only one motif, Motif-2, was found. The conserved motif composition and gene structure of the same subfamily were similar, thus verifying the reliability of the phylogenetic tree classification.

Chromosome distribution and duplication of SmAP2/ERF superfamily genes

The chromosome location of the identified SmAP2/ERF genes was constructed using TBtools. In total, 310 genes from the AP2/ERF superfamily were unevenly distributed on 38 chromosomes (Fig. 4); 54 other genes located on scaffolds were not illustrated in Fig. 4. The chromosome with the largest number of AP2/ERF genes was Chr21, which had 22 genes. Only one AP2/ERF gene each was located on Chr14 and Chr36. On the four chromosomes Chr1, Chr3, Chr22, and Chr27, only two AP2/ERF genes were found. In 38 chromosomes, most of the AP2/ERF genes from different subgroups were arbitrarily distributed, such as five of six RAV genes located on Chr15, Chr37, Chr34, Chr31, and Chr11. Moreover, the two Soloist genes were distributed on Chr29 and Chr5. However, SmERF B3 subgroup members clustered together with three genes as a cluster unit. We found 12 clusters in 12 chromosomes (Fig. 4), which accounted for 62% of the whole SmERF B3 subgroup.

Figure 4 Schematic representations for the chromosomal distribution and tandem duplication of SmAP2/ERF genes.

The red lines indicate tandem duplicated AP2/ERF gene pairs, which are indicated in green. The SmERF-B3 subgroup members labeled with blue clustered on the same chromosome. The chromosome number is indicated to the left of each chromosome.

In addition, we also analyzed the tandem duplication events (TDs) of the AP2/ERF genes located within in the 200-kb range of chromosomal regions of the Salix matsudana genome. Eleven TD regions, which included 23 SmAP2/ERF genes, clustered into 11 linkage groups (LGs) of the Salix matsudana genome (Fig. 4). LGs that contained cluster repeat genes were Chr7, Chr8, Chr10, Chr13, Chr17, Chr19, Chr21, Chr24, Chr28, Chr31, and Chr33. All genes of the repeat clusters were localized within a genomic segment of less than 20 Kb; for example, SmDREB A4-20 and SmDREB A4-19 were localized on a 3.6-Kb segment of Chr24. On Chr8, three genes clusters (SmERF B3-6, SmERF B3-7, and SmERF B3-8) located on a less than 12-Kb segment. In 11 tandem repeats, six came from the SmERF B3 subgroup, two came from the SmDREB A4 subgroup, and one each came from the SmDREB A1 and AP2 subgroups. SmERF B3-40 and SmERF B3-39 tandem repeat pairs had 97% protein sequence identity.

In addition to tandem duplications, many segmental duplication events (SDs) were found in Salix matsudana by MCScanX (Fig. 5, Table S5). We found a total of 28,348 collinear gene pairs (not shown) in the Salix matsudana genome, from which 298 AP2/ERF collinear gene pairs were identified. Then, Ka, Ks, and Ka/Ks ratios of these 298 AP2/ERF collinear gene pairs were calculated to estimate the divergence time (T value) and selection pressure among duplicated SmAP2/ERF gene pairs (Table S6). All of the Ka/Ks values were below 1, which indicated that these genes might have experienced strong purifying selective pressure during evolution. Among the 298 AP2/ERF collinear gene pairs, 198 were located on duplicated segments on 38 chromosomes in Salix matsudana (Fig. 5 and Table S3). The collinear gene pairs in the Salix matsudana genome were visualized by Circos, and the gene pairs were linked by lines (grey lines indicated all gene pairs, red lines indicated AP2/ERF collinear gene pairs).

Figure 5 Schematic representations of the segmental duplication and interchromosomal relationships of SmAP2/ERF genes.

Grey lines indicate all syntenic gene pairs in the Salix matsudana genome, red lines indicate syntenic relationships between SmAP2/ERF genes. The orange color columes outside of the circle indicated the gene density on each 38 chromosomes. The deeper color means the higher density of genes.

Synteny analysis of AP2/ERF genes between Salix matsudana and two related Salicaceae species, Populus trichocarpa and Salix purpurea

To further infer the phylogenetic mechanisms of the SmAP2/ERF family, we constructed two comparative syntenic maps of Salix matsudana with two related species, Populus trichocarpa (Fig. 6A) and Salix purpurea (Fig. 6B). Collinear AP2/ERF genes pairs between Salix matsudana and other two species are listed in Table S7. A total of 263 SmAP2/ERF genes showed syntenic relationships with 183 genes from Populus trichocarpa, and 248 SmAP2/ERF genes showed syntenic relationships with 144 genes from Salix purpurea. The number of orthologous pairs between Salix matsudana and Populus trichocarpa, and Salix matsudana and Salix purpurea were 423 and 292, respectively (Table S7). Some PtAP2/ERF and SpAP2/ERF genes were found to be associated with at least four syntenic gene pairs. Interestingly, the number of collinear gene pairs identified between Salix matsudana and Salix purpurea were less than that between Salix matsudana and Populus trichocarpa.

Figure 6 Synteny analysis of AP2/ERF genes between Salix matsudana and two related Salicaceae species, Populus trichocarpa and Salix purpurea.

(A) Synteny analysis of AP2 genes between Salix matsudana and Populus trichocarpa. (B) Synteny analysis of AP2 genes between Salix matsudana and Salix purpurea. Gray lines in the background indicate the collinear blocks within Salix matsudana and other plant genomes, whereas red lines highlight syntenic AP2/ERF gene pairs.

In the comparative syntenic map between Salix matsudana and Populus trichocarpa, syntenic links were found between all 19 Populus trichocarpa chromosomes and all 38 Salix matsudana chromosomes (Fig. 6A). Alternatively, in the comparative syntenic map between Salix matsudana and Salix purpurea, there were no syntenic links between Chr1, Chr12, and Chr36 from Salix matsudana, and Chr15Z and Chr15W from Salix purpurea (Fig. 6B).

Specific expression of AP2/ERF superfamily genes under salt stress

To investigate the physiological roles of SmAP2/ERF genes in salt stress tolerance, we identified the expression patterns of SmAP2, SmERF, and SmDREB subgroup genes from the RNA sequencing data. The RNA transcripts of 285 genes were identified from the RNA sequencing data. Using fold change ≥ 2 and false discovery rate (FDR) <0.01 as screening criteria, 68 SmAP2/ERF genes were identified as DEGs. The DEGs names were listed in Table S8. By inputting the FPKM values (Fragments Per Kilobase of transcript per Million fragments mapped) of these genes in TBtools, three heatmaps were constructed using Log10-transformed values of the FPKM values to demonstrate the expression pattern change under salt stress (Fig. 7).

Figure 7 Hierarchical clustering of AP2 genes and heatmap that demonstrates the differential expression patterns of SmAP2/ERF genes in roots before and after salt stress.

The Log_10-transformed expression values of the FPKM values of 12 samples were used to draw the heatmap. The color bar indicates the gene expression level. (A) Heatmap and hierarchical clustering representation of 47 AP2 members. (B) Heatmap and hierarchical clustering representation of 108 DREB members. (C) Heatmap and hierarchical clustering representation of 130 ERF members. Sample1-1/Sample1-2/Sample1-3 means three replicated experiments of YJ CK, the ‘Yanjiang’ roots treated with water; Sample1N-1/Sample1N-2/Sample1N-3 means three replicated experiments of YJ NT, the ‘Yanjiang’ roots treated with 150 mM NaCl; Sample2-1/Sample2-2/Sample2-3 means three replicated experiments of 9901 CK, ‘9901’ roots treated with water; and Sample2N-1/Sample2N-2/Sample2N-3, means three replicated experiments of 9901 NT, ‘9901’ roots treated with 150 mM NaCl. The asterisks idictate the DEGs.

The expression patterns of 285 genes are illustrated in Fig. 7, and included 47 AP2 (Fig. 7A), 108 DREB (Fig. 7B), and 130 ERF subgroup genes (Fig. 7C). In the AP2 subgroup, the Log10-transformed values of 31 genes were <3, which indicated lower expression in the root and no response to salt stress. Five DEGs had differential expression patterns. The expression levels of four genes (SmAP2-38, SmAP2-4, SmAP2-3, and SmAP2-33) were induced by salt stress, whereas the expression of gene SmAP2-15 decreased after salt stress. In the DREB subgroup, 108 genes including 27 DEGs were present in the heatmap, and expression levels of 10 genes, such as SmDREB A1-10, SmDREB A1-9, and SmDREB A1-7, were highly induced by salt stress and remained higher. In the ERF subgroup heatmap, 130 genes (containing 33 DEGs) were included. The expression levels of 13 genes were intensely upregulated by salt stress, but only the expression of SmERF B4-1 was higher. Three genes were downregulated by salt stress, including SmERF B3-52. In many paralog gene pairs, we found one gene with higher expression, whereas the other gene had lower expression, such as SmDREB-A9/SmDREB-A10, SmAP2-33/SmAP2-39 and SmERF-9/SmERF-10 gene pairs. Fourteen genes with upregulated expression patterns were verified by qRT-PCR (Real-time Quantitative PCR) (Fig. 8). From the results, we found that most genes’ expression patterns were consistent with the FRKM values except SmERF B3-42 and SmAP2-15 genes. Both in ‘Yanjiang’ and ‘9901’ samples, the expression level of thirteen genes was induced after salt treatment, but the expression level of seven genes (SmAP2-33, SmDREBA4-24, SmDREBA1-4, SmDREBA1-7, SmDREBA5-23, SmERF B3-45 and SmERF B4-1) was much higher induced in ‘9901’ than that in ‘Yanjiang’, which was not found in RNA sequencing results.

Figure 8 Verification of the SmAP2/ERF genes with differentially expressed patterns under salt stress by quantitative real-time PCR.

For salt stress, ‘Yanjiang’ and ‘9901’ roots that were subjected to 20 days of hydroponic culture and then treated with 150 mM sodium chloride for 4 h. The control was an untreated Yanjiang sample. Three biological replicates for each sample were performed, and bars represent the standard deviations of the mean. Asterisks on top of the bars indicate statistically significant differences between stress treatment and the control (*0.01 < P < 0.05; ** p < 0.01, Student’s t-test). Gene expression profiles were evaluated using the 2−ΔΔCt method, and the control value was normalized to 1.

Discussion

Salix is one of the few woody plants with a large number of polyploid taxa, in S. matsudana, both tetraploid and diploid individuals have been observed (Guo et al., 2016). In our previous experiment, we sequenced the Salix matsudana, an allotetraploid salix. Tetraploid Salix is valuable because they have higher tolerance to abiotic stress than their diploid relatives (Guo et al., 2016); therefore, they can be planted beachside to alleviate soil salinity and improve the ecological environment (Zhang et al., 2016). The molecular mechanism of salinity response regulation is very complex, and AP2/ERF TFs are key regulators in plants (Xie et al., 2019). Here, we identified 364 AP2/ERF gene members in Salix matsudana, and characterized their classification, chromosome location, gene structure, and syntenic relationships of these genes within the genome and between other species. We also revealed the expression patterns under salt stress. These efforts can serve as a first step in comprehensive functional characterization of AP2/ERF genes by reverse genetic approaches and molecular genetics research.

As an allotetraploid species, Salix matsudana has more AP2/ERF gene members than other plants selected for comparison, including three Salicaceae family relatives (Table S3). The total number of genes is approximately double compared with poplar and two willow relatives, but the proportions of some subgroups were slightly different. Salix arbutifolia had a higher percentage (50.8% >45%) of ERF subfamily members, but a lower proportion (33% <38%) of DREB subfamily members compared with other species (Table S3). For DREB-A1 and ERF-B2 subgroups, the highest percentage was found in Salix purpurea, and there were the same number of or more members of these two subgroups compared with other species, including the tetraploid Salix matsudana. For ERF-B3, Salix arbutifolia had the highest subgroup percentage (18.5%). In Salix matsudana, ERF-B6 had the lowest percentage (5.7%), whereas the subfamily AP2 had the highest percentage (15.1%). These data indicated that, during evolution, AP2/ERF family subgroup members probably underwent gene duplication or loss and therefore evolved into the specific AP2/ERF subgroup proportions in each species.

A phylogenetic tree that included 364 genes from Salix matsudana and 48 genes from A. thaliana and Populus trichocarpa was constructed (Fig. 1). All subgroups were clustered together. Eight genes with one AP2 domain were classified into the AP2 family because of a close phylogenetic relationship. This classification was similar to that in Arabidopsis, in which four genes involved in the AP2 family contained a single AP2 domain (Nakano et al., 2006).

The gene intron/exon structure and conserved motifs were identified in the 364 SmAP2/ERF members. Similar to that of the AP2/ERF genes from other species, such as cauliflower and radish, the AP2 subfamily had more introns and the ERF subfamily had fewer (Karanja et al., 2019; Li et al., 2017a). Previous studies found that intron number and distribution are related to plant evolution, and introns of the ERF family genes were probably lost during evolution in higher plants (Tang et al., 2016; Zhang & Li, 2018). In total, 215 of the 301 members (70%) of the ERF family had no introns, which was a little less than that of tartary buckwheat (Liu et al., 2019) and also consistent with previous findings (Li et al., 2017a; Li et al., 2019a).

Through the conserved domains and motifs, TFs play roles in gene expression regulation by promoter binding, transcription activation, and protein–protein interactions (Liu, White & Macrae, 2010). Motif analysis showed that Motif-6, Motif-8, and Motif-10 were specifically detected in different groups of the AP2/ERF subfamily; seven other motifs were all related to the AP2 domain (Fig. 3B). Motif-8 was specifically found in the DREB subgroup, such as in the DREB-A1, DREB-A4, and DREB-A5 clades. Motif-10 was mostly found distributed on proteins from the ERF-B3, DREB-A2, and DREB-A4 clades. Motif-6 was specifically located between the two AP2 domains of AP2 subgroup members. These results indicate that, although some motifs of the AP2/ERF family genes were highly conserved and involved in DNA binding, such as motifs from the AP2 domain, the functions of other subgroup-specific motifs are still unknown, and more work is required to clarify their regulatory functions.

Based on the genome assembly data, 301 genes were anchored on the 38 chromosomes (LGs), but they were unevenly distributed. Eleven TDs were found on 11 chromosomes, and seven tandem duplication gene pairs came from the SmERF B3 subgroup, which included three duplicated genes (SmERF B3-6, SmERF B3-7, and SmERF B3-8) that clustered together. Apart from the tandem duplication cluster, SmERF B3 members typically clustered on a chromosome, with three genes as a unit. In 12 clusters, 37 SmERF B3 genes were found. This phenomenon were also found in Populus trichocarpa, thirteen PtERF B3 genes located in 4 clusters, which indicated that in the evolution of Salix matsudana, apart from the chromosome duplication, segmental duplication were also happened.

Using MCScanX, we found a total of 28,348 collinear gene pairs in the Salix matsudana genome, from which 299 AP2/ERF collinear gene pairs were identified; this indicated that, during evolution, the Salix matsudana genome experienced a whole genome duplication event. Population genetic theory predicts that, after duplication, some redundant duplicate copies will be silenced and eliminated, and other retained paralogs will obtain sub- or neofunction by DNA mutation in coding or regulatory sequences (Adams et al., 2003; Hou et al., 2019; He & Zhang, 2005).

Then, we calculated Ka, Ks, and Ka/Ks ratios of these 298 AP2/ERF collinear gene pairs to estimate the divergence time and selection pressure. All Ka/Ks values were below 1, which indicated that these genes might have experienced strong purifying selective pressure during evolution. It was previously reported that purifying selection would lead to the loss of redundant genes (Kondrashov et al., 2002). Based on the gene number of most subgroups, we did not find any obvious evidence of gene loss, but in the RAV subgroup, there was an exception; there were only six members in Salix matsudana, which is identical to the gene number in Arabidopsis. Based on the gene loss hypothesis, the duplication paralogs of RAVs may have been lost during genome evolution because of their rapid evolutionary rate.

Approximately 52–59 million years ago (Mya), willow and poplar, which are two modern taxa, originated from a diploid progenitor, but when and how Salix matsudana originated and experienced chromosome duplication remains largely unknown (Hou et al., 2016). The allotetraploid Salix matsudana may originated from hybridization between two diploid salix germplam and subsequently genome duplication. In our study, the divergence time (T Value) of gene pairs can be classified mainly into two groups, a group and b group with two time periods, 2–8 Mya (average value, 5 Mya) and 20–36 Mya (average value, 26 Mya) respectively (Fig. S3; Table S6). Gene pairs with a divergence time of 20–36 Mya were probably paralogs from two diploid hybrid parents; whereas 2–8 Mya is probably the divergence time of paralogs after whole genome duplication events. These data indicated that there were two large gene duplication events, 8 and 36 Mya ago in the AP2 / ERF family in the evolution of Salix matsudana.

Similar to the findings of a previous report, alignment of a Salix linkage map to the Populus genomic sequence revealed macrosynteny between willow and poplar genomes (Hanley, Mallott & Karp, 2006) (Figs. 6A, 6B). Synteny analysis of Salix matsudana vs Populus trichocarpa, and Salix matsudana vs Salix purpurea revealed 423 and 292 orthologous pairs, respectively. In total, 263 SmAP2/ERF genes had syntenic relationships with 183 genes in Populus trichocarpa, whereas 248 SmAP2/ERF genes showed syntenic relationships with 144 genes in Salix purpurea. Interestingly, the collinear gene pairs identified between Salix matsudana and Salix purpurea were less than that from Salix matsudana and Populus trichocarpa. Syntenic links were found between all 19 Populus trichocarpa chromosomes and all 38 Salix matsudana chromosomes, but there were no syntenic links between Chr1, Chr12, and Chr36 from Salix matsudana and Chr15Z and Chr15W from Salix purpurea. Salix has 300–500 species and considerable variation, ranging from shrubs to trees (Argus, 1997); willow may evolve faster, which would lead to them being more diverse. Researchers proposed that Populus might be evolutionarily more primitive than Salix (Dai et al., 2014; Hou et al., 2019). From our results, we could infer the evolutionary relationships of three Salicaceae species (Populus trichocarpa, Salix matsudana, and Salix purpurea); Populus trichocarpa was the most primitive taxon, Salix purpurea was the most derived taxon, and Salix matsudana was located between them but was genetically more closely related to Populus trichocarpa than Salix purpurea.

Plants must adapt to various biotic and abiotic stresses because they are immobile in their life cycles. For example, Salix matsudana must adapt to the soil salinity when grown along coastal beaches. Consequently, some AP2/ERF TFs play important roles in plants by facilitating defense against stress and improving resistance. From the RNA sequencing data, the DEGs of SmAP2/ERF were identified and the expression heatmaps were presented to show the expression patterns under salt stress (Fig. 7). The expression levels of four genes from the AP2 subgroup, 10 genes from the DREB subgroup, and 13 genes from the ERF subgroup were strongly induced by salt stress, but only the expression levels of four genes were downregulated after salt stress. Most of genes’ expression patterns were verified by qRT-PCR. The expression pattern of many AP2/ERF gene pairs with evolutionary relationships differed, which indicated that the AP2/ERF gene family may have changed at the transcriptional regulation level following polyploidization. That finding provides additional evidence that redundant duplicated gene pairs experienced functional divergence based on expression pattern change. These differentially expressed SmAP2/ERF genes could be selected as candidate genes; such as SmDREB A1-4 and SmERF B3-45, further exploration on their roles under salt stress will reveal molecular mechanisms responsible for salinity stress responses in Salix matsudana.

In conclusion, 364 AP2/ERF TFs were identified in Salix matsudana. Clustering and phylogenetic analysis were conducted to classify these TFs into 15 subgroups. Chromosome location, gene structure, and conserved motifs were identified for 364 AP2/ERF TFs. Evolutionary relationships of these genes were revealed by tandem and segmental duplication gene pair identification, divergence time estimation, and T value calculation, which indicated that the progenitor of Salix matsudana, two diploid salix germplasms, underwent hybridization and genome duplication not more than 10 Mya. Synteny analysis with other species showed macrosynteny between willow and poplar AP2/ERF genes, and Salix matsudana was genetically more closely related to Populus trichocarpa than Salix purpurea. The AP2/ERF TFs were also confirmed to exhibit differential expression patterns during salt stress. The functions of these genes should be investigated in future studies to better clarify the mechanism of salt tolerance regulation in Salix matsudana, which will be helpful for breeders in salt tolerance varieties selection.

Conclusion

In this study, 364 SmAP2/ERF genes of Salix matsudana were identified and renamed according to the chromosomal location of the SmAP2/ERF genes. Gene classification, gene structure and conserved motifs were analyzed in detail. Investigation results on syntenic relationships between the SmAP2/ERF genes and AP2/ERF genes from other species elucidated that the progenitors of Salix matsudana underwent whole genome duplication not more than 10 Mya and Salix matsudana is genetically more closely related to Populus trichocarpa than to Salix purpurea. Moreover, analyses on the differential expression patterns of SmAP2/ERF genes during salt stress can help to reveal the mechanism of salt tolerance regulation in Salix matsudana.

Supplemental Information

Supplemental Information 1 Supplemental Tables

Click here for additional data file.

Supplemental Information 2 Alignment of 412 AP2 domains

To better classify these SmAP2 genes, 48 AP2 domains from known categories of Arabidopsis and Populus trichocarpa AP2 genes were selected to carry out multiple sequence alignment with AP2 domains of SmAP2/ERF proteins using ClustalW in Website https://npsa-prabi.ibcp.fr/cgi-bin/npsa_automat.pl?page=/NPSA/npsa_clustalw.html.

Click here for additional data file.

Supplemental Information 3 Conserved motifs in proteins of SmAP2/ERF superfamily

The online tool MEME (http://meme-suite.org/tools/meme) was used to search for conserved motifs of SmAP2/ERF superfamily proteins. The optimized parameters were employed as follows: any number of repetitions, maximum number of motifs = 10, and the optimum width of each motif was 6–50 residues.

Click here for additional data file.

Supplemental Information 4 The divergence time (T Value) of gene pairs from two groups

The divergence time (T Value) of gene pairs can be classified mainly into two groups, a group and b group with two time period, 2–8 Mya (average value, 5Mya ) and 20–36 Mya (average value, 26Mya) respectively.

Click here for additional data file.

Supplemental Information 5 Two SmAP2_domain hmm profiles and all SmAP2/ERF gene CDS sequences

Click here for additional data file.

Supplemental Information 6 All gene fpkm value of 12 samples

RNA Sequencing data.

Click here for additional data file.

Supplemental Information 7 The original hmmsearch–domtblout results

We downloaded the Hidden Markov Model (HMM) profile for the AP2/ERF TFs from the Pfam database (http://pfam.xfam.org/) with Pfam accession number PF00847 as the search keyword. An alternative HMM profile was built by sequence alignment using ClustalW. Using an in-house Perl script with two HMM profiles as queries, hmmsearch was carried out by searching the Salix matsudana protein databases with default parameters.

Click here for additional data file.

Supplemental Information 8 Raw data for QRT-PCR

The data exported from ABI 7500 software v2.3.

Click here for additional data file.

Supplemental Information 9 S. matsudana genome annotation and protein data

Click here for additional data file.

We thank Mallory Eckstut, PhD, from Liwen Bianji, Edanz Editing China for editing the English text of a draft of this manuscript.

Abbreviations

AP2/ERF AP2-like ethylene-responsive transcription factor

FPKM Fragments Per Kilobase of transcript per Million fragments mapped

HMM Hidden Markov Model

Ka Nonsynonymous substitution rate

Ks Synonymous substitution rate

Mya Million years ago

qRT-PCR Real-time Quantitative PCR

SDs Segmental duplication events

TDs Tandem duplication events

TFs Transcription factors

Additional Information and Declarations

Competing Interests

Author Contributions

Data Availability

The authors declare there are no competing interests.

Jian Zhang and Shi zheng Shi conceived and designed the experiments, performed the experiments, analyzed the data, prepared figures and/or tables, authored or reviewed drafts of the paper, and approved the final draft.

conceived and designed the experiments, performed the experiments, analyzed the data, prepared figures and/or tables, authored or reviewed drafts of the paper, and approved the final draft.

Yuna Jiang performed the experiments, analyzed the data, authored or reviewed drafts of the paper, and approved the final draft.

Fei Zhong, Guoyuan Liu, Chunmei Yu and Bolin Lian performed the experiments, authored or reviewed drafts of the paper, and approved the final draft.

Yanhong Chen conceived and designed the experiments, analyzed the data, prepared figures and/or tables, authored or reviewed drafts of the paper, and approved the final draft.

The following information was supplied regarding data availability:

Raw RNA sequencing data is available at the CNGB Sequence Archive (CNSA) of China National GeneBank DataBase (CNGBdb): CNP0001274.

Raw data, including the FPKM values of all genes from RNA sequencing, are available in the Supplementary Files.

The latest S. matsudana genome sequences are available at NCBI: PRJNA687297.

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
