# Peer review of "Genome-wide investigation of the AP2/ERF superfamily and their expression under salt stress in Chinese willow (Salix matsudana)"

_PeerJ, doi:10.7717/peerj.11076_

## Round 0.1 · original submission · Major Revisions

Please revise your manuscript according to reviewers' comments.

Reviewer 1 ·

Basic reporting

no comment

Experimental design

no comment

Validity of the findings

no comment

Additional comments

In this study, the authors characterized of the AP2/ERF transcription factors gene family within tetraploid tree species, Salix matsudana. They analyzed the bioinformatic results of AP2/ERF gene family, including chromosomal location, conserved motif, phylogenetic tree, gene structure, gene number, syntenic analysis etc. Further, expression pattern of SmAP2/ERF genes under salt stress were screened to identify the salt-tolerant candidate genes in willow. Some concerns are listed below:

1. The author identified 364 AP2/ERF TFs genes in total, while they showed the expression patterns of 285 genes in figure 7, why? Actually, they screened 31 DEGs AP2/ERF TFs genes in response to the salt stress by RNA-seq, however, they further verified 14 genes by qRT-PCR, which was inconsistent. In addition, in my opinion, the author only need present the expression levels of 31 DEGs in fig.7, other information could be moved to the supplementary file. Finally, what the authors mean by writing Sample 1, Sample 1N, Sample 2, Sample 2N in figure 7? What’ s their relationship with ‘Yanjiang’ and ‘9901’.
2. For Figure 3, the phylogenetic tree was not that robust as showing low bootstrap values, I advise this figure could be moved to the supplementary file.
3. Line 226, Using TopHat2 software, the clean reads were mapped to the reference genome sequence of S. matsudana (Kim et al., 2013). This citation need be placed after to the TopHat2 software. Otherwise, readers would mistake it for the reference genome.
4. Where did the author deposit the clean data of RNA-seq.

·

Basic reporting

Please double check the website (https://js-568garden.cn/?list_6/99.html) to make sure that data are available for public.

Experimental design

no comment

Validity of the findings

no comment

Additional comments

This manuscript concerns AP2/ERF gene family and their potential roles in salt stress in Salix matsudana, a tetraploid ornamental tree species. The manuscript was well organized, and the content was qualified for publication in PeerJ. Before the final acceptance, there are some minor errors require corrections, as listed below:
(1) Gene names should be in italic type (e.g. Line 98, Line 99, Line 101, Line 155, and ……);
(2) Species names should be in italic type (e.g. Line 133, Line 134, Line 155, and ……);
(3) Figure legend for Fig. 3B "Motif composition of tartary buckwheat AP2/ERF proteins" should be revised.
(4) More details are highlighted in the attached file.

Reviewer 3 ·

Basic reporting

The manuscript is an computacional analysis of AP2/ERF transcriptional factors in Salix matsudana compared to Salix purpurea and Populus trichocarpa also shows RNA expression of some genes related to salt tolerance in S. matsudana.
The work is well written, results are quite well shown and add knowledge to this tree species with importance in China.

Experimental design

I have some comments and questions to the authors:
Line 104: correct the word alfalfa
Line 114: Koidz should not be italic
Line 151: what is the meaning of "mimic RNA sequencing"
Line 164: please review the Pfam code (PF00847.16) and add the gathering cut-off value.

Line 171: why did your used SMART ? This tool also use Hidden Markov Model.

Line 179, for multiple sequence alignment and phylogenetic tree, did you used dataset of pfam or your "in-house Perl script method? Could you mention, why they did not use the proteins selected by Pfam?.

Line 181: what was the criteria to select only the 48 AP2 domains of Arabidopsis and Populus, please, could you increase the citations of these works.

Line 190: the Salix matsudana genome is already assembled? and annotaded?, please if so, you could add the url.

Line 201: The citation should be just after TBtools.

Line 216: how the gene pairs were estimated for the nonsynonymous substitution rate (Ka) and Synonymous substitution rate (Ks)?, it was all vs all? are they paralogs?.

Line 218: I dont find the formula T = Ks/2λ in the Lynch study, and the The clock-like rate λ value for Populus, please review.

Line 228: the fragments per kilobase of transcript per million fragments mapped (FPKM) was estimated by your team? if yes, could you mention how it was calculated, if it was not you could you please add the citation.

Line 237: I would like to suggest to add the supplementary material with .domtblout information of hmmsacan, to make the search and the result of that search easier.

Line 244: The criteria used to cladificate the ERF and AP2 subfamilies, were according to the presence of the WLG and YLG elements respectively?, please clarify this point, including also the same criteria for RAVE and Soloist subgroups.

Line 248: Please change "Basing" to Based on

Line 299: The S4 supplementary data is missing. Also could you add the the bioinformatic tool reference used to determine FPKM?

Line 344: How to explain the large variations in the divergence times of the genes (Table S3), do you know if there were ploidization events in the species, do you consider that these data correlate with whole genome duplication?.

Line 399: The statement " Tetraploid Salix is valuable because they have higher tolerance to abiotic stress than their diploid relatives" is conclusion or you have a reference that confirm that?

Line 461: I would like to suggest adding a histogram, presenting the Ks / Ka or Ks vs frequency or density to better visualizing that result that I find interesting.

Line 474-478: Since the data indicated that the two diploid progenitors of Salix matsudana underwent hybridization and genome duplication not more than 10 Mya with only one gene family? I think it would be more appropriate to mention that apparently there were two large gene duplication events 8 and 36 Mya ago in the AP2 / ERF family.


Figure 1, please add the MEGA 7.0 citation
Figure 3, correct the word "members" . The protein length of figure b and c do not have the same scale, please check
Figure 7, please, clearly indicate the samples with the two conditions (before and after salt stress)

Validity of the findings

The results and discussion of this article add knowledge Salix species related to stress tolerance.

---

## Round 0.2 · Minor Revisions

The reviewers recommended minor revision.

Reviewer 1 ·

Basic reporting

no comment

Experimental design

no comment

Validity of the findings

no comment

Additional comments

The author have addressed most of the comments, however, For Figure 3, the author should use other method or software to perform the phylogenetic analysis, to make the tree robust.

Reviewer 3 ·

Basic reporting

no comment

Experimental design

I consider the article was improved. There were some questions not answered from my previous review but the authors make changes in the manuscript.

Validity of the findings

no comment

Additional comments

The authors have improved the article.

---

## Round 0.3 · Minor Revisions

The section editor has made comments on your manuscript. Please revise it accordingly.

"1) RNAseq data needs to be deposited into public repository; they have only deposited the genome.

2) Differentially expressed genes are discussed in the paragraph starting on line 377 but no FDR values, etc are presented. The methods describe using DEseq to determine which genes are differentially expressed but this analysis seems to not be in the paper. DEseq should be used to determine which genes are differentially expressed.

3) DEseq requires input of raw counts, not FPKM."

Reviewer 1 ·

Basic reporting

no comment

Experimental design

no comment

Validity of the findings

no comment

Additional comments

The authors have addressed all of my comments and I have no further suggestions.

---

## Round 0.4 · accepted · Accept

The revised version is acceptable.